EMBO
Molecular Medicine

# Monocytopenia, monocyte morphological anomalies and hyperinflammation characterise severe COVID-19 in type 2 diabetes

Fawaz Alzaid[1,*,†] (iD), Jean-Baptiste Julla[1,2,†], Marc Diedisheim[1,3], Charline Potier[1], Louis Potier[1,4], Gilberto Velho[1], Bénédicte Gaborit[5,6], Philippe Manivet[7], Stéphane Germain[8], Tiphaine Vidal-Trecan[2], Ronan Roussel[1,4], Jean-Pierre Riveline[1,2], Elise Dalmas[1], Nicolas Venteclef[1] & Jean-François Gautier[1,2,**] (iD)

## Abstract

Early in the COVID-19 pandemic, type 2 diabetes (T2D) was marked as a risk factor for severe disease and mortality. Inflammation is central to the aetiology of both conditions where variations in immune responses can mitigate or aggravate disease course. Identifying at-risk groups based on immunoinflammatory signatures is valuable in directing personalised care and developing potential targets for precision therapy. This observational study characterised immunophenotypic variation associated with COVID-19 severity in T2D. Broad-spectrum immunophenotyping quantified 15 leucocyte populations in peripheral circulation from a cohort of 45 hospitalised COVID-19 patients with and without T2D. Lymphocytopenia and specific loss of cytotoxic CD8+ lymphocytes were associated with severe COVID-19 and requirement for intensive care in both non-diabetic and T2D patients. A morphological anomaly of increased monocyte size and monocytopenia restricted to classical CD14$^{Hi}$ CD16$^{-}$ monocytes was specifically associated with severe COVID-19 in patients with T2D requiring intensive care. Increased expression of inflammatory markers reminiscent of the type 1 interferon pathway (IL6, IL8, CCL2, INFB1) underlaid the immunophenotype associated with T2D. These immunophenotypic and hyperinflammatory changes may contribute to increased voracity of COVID-19 in T2D. These findings allow precise identification of T2D patients with severe COVID-19 as well as provide evidence that the type 1 interferon pathway may be an actionable therapeutic target for future studies.

**Keywords** COVID-19; inflammation; monocyte; SARS-CoV-2; type 2 diabetes

**Subject Categories** Immunology; Metabolism; Microbiology, Virology & Host Pathogen Interaction

## Introduction

Coronavirus disease 19 (COVID-19), an infectious disease caused by the severe acute respiratory syndrome coronavirus (SARS-CoV)-2, was first identified in Wuhan, China, in December 2019. SARS-CoV-2 induces strong systemic inflammation and acute injury of the lung and potentially other organs (Merad & Martin, 2020). As of 30 June 2020, over 10 million cases have been reported worldwide, with an estimated mortality of 5–6% (Dong *et al*, 2020). Early reports stated that individuals with diabetes are at high risk of complications or death from severe COVID-19 (Zhu *et al*, 2020). Later studies indicated that factors such as age, obesity or hypertension, although commonly associated with diabetes, independently increase risk (Wu *et al*, 2020; Zhou *et al*, 2020). The complex aetiologies of these conditions and of type 2 diabetes (T2D) make deciphering the mechanisms that increase severity of COVID-19 a complex task.

Recent reports have found important links between systemic metabolism, glucose homeostasis and responses to COVID-19. Notably, studies have shown that glycaemic variability strongly influences outcome in COVID-19, where poorly controlled blood glucose was associated with markedly higher mortality compared to patients with well-controlled blood glucose (Zhu *et al*, 2020).

1   Cordeliers Research Centre, INSERM, IMMEDIAB Laboratory, Sorbonne Université, Université de Paris, Paris, France
2   Department of Diabetes, Clinical Investigation Centre (CIC-9504), Lariboisière Hospital, Assistance Publique – Hôpitaux de Paris, Paris, France
3   Department of Diabetology, Cochin Hospital, Assistance Publique Hôpitaux de Paris, Université de Paris, Paris, France
4   Department of Diabetology, Endocrinology and Nutrition, Bichat Hospital, Assistance Publique - Hôpitaux de Paris, Paris, France
5   INSERM, INRA, Aix Marseille University, C2VN, Marseille, France
6   Endocrinology, Metabolic Diseases and Nutrition Department, Assistance Publique Hôpitaux de Marseille, Marseille, France
7   Centre de Ressources Biologique "biobank Lariboisière", BB-0033-00064, APHP, Nord, Université de Paris, Paris Diderot, Hôpital Lariboisière, Paris, France
8   Center for Interdisciplinary Research in Biology (CIRB), College de France – Centre National de la Recherche Scientifique (CNRS), Institut National de la Santé et de la Recherche Médicale (INSERM), Paris Sciences et Lettres (PSL) Research University, Paris, France
    *Corresponding author (lead contact). Tel: +33 1 44 27 81 08; E-mail: fawaz.alzaid@gmail.com
    **Corresponding author. Tel: +33 1 44 27 81 08; E-mail: jean-françois.gautier@aphp.fr
    †These authors contributed equally to this work as first authors

Similarly, high glycosylated haemoglobin (HbA1c), a proxy of glycaemic instability, has been associated with low oxygen saturation, inflammation and hypercoagulability in patients with COVID-19 (Wang *et al*, 2020b). More recently, mechanistic studies have shown that the elevated glucose that sustains inflammatory metabolism in immune cells directly promotes viral replication and cytokine production in SARS-CoV-2 infection (Codo *et al*, 2020). Other potential immune-mediated mechanisms increasing severity of COVID-19 in patients with T2D include (i) higher affinity cellular binding and efficient virus entry, (ii) decreased viral clearance and (iii) altered immune responses leading to hyperinflammation and aggravated cytokine storm syndrome.

Deregulation of the viral immune response in T2D is a plausible hypothesis. Lymphopenia has been consistently reported as one of the most frequent immune abnormalities in Chinese populations suffering severe COVID-19 (Tan *et al*, 2020). Peripheral lymphocyte counts are low, with a higher proportion of pro-inflammatory Th17 CD4$^+$ T cells, as well as elevated cytokine levels. In addition to altered T-cell dynamics, studies revealed altered monocyte phenotypes in severe forms of COVID-19 (Zhang *et al*, 2020). The important implication of the innate immune system in COVID-19 aetiology is also evidenced by aberrant macrophage activation at the site of infection and in peripheral tissues (Park, 2020).

Previous studies in T2D have highlighted abnormalities in monocyte phenotypic transition states related to disease severity and cardiovascular risk (Rogacev *et al*, 2012; Menart-Houtermans *et al*, 2014). Interactions between the heightened immunoinflammatory states in T2D and in COVID-19 cases may well be at the root of increased susceptibility of T2D patients to severe COVID-19. Deregulation of T-cell or monocyte frequency and function may result in unchecked innate and adaptive immune responses. Such responses lead to the uncontrolled and sustained inflammation observed in severe COVID-19 and at higher risk of mortality.

Herein, the immunophenotypic profile of diabetic and non-diabetic COVID-19 patients revealed specific phenotypic and morphological alteration of monocytes in T2D patients. Interestingly, the type 1 interferon sensitive transcription factor, the interferon regulatory factor (IRF)-5, is associated with altered monocyte fate in T2D. Finally, we provide evidence that a loss of CD8$^+$ T cells and classical monocytes concomitant to increased IRF5 expression dictate severity of COVID-19 in T2D patients.

## Results

### Type 2 diabetes is associated with decreased monocyte frequency and phenotypic alterations in COVID-19 patients

To characterise the influence of pre-existing diabetes in COVID-19 patients, we recruited a cohort of 45 COVID-19 patients from Lariboisière Hospital's University Centre for Diabetes and its Complications including 30 patients with T2D and 15 non-diabetic (ND) patients. When comparing T2D to ND patients, we have no differences in clinical or demographic criteria, with the exception of increased HbA1c and hypertension in T2D relative to ND (Table 1). HbA1c % was fourfold higher, and prevalence of hypertension was 1.3-fold higher in T2D with COVID-19 relative to ND patients with COVID-19.

**Table 1. Characteristics of COVID-19 patients in the non-diabetic (ND) and type 2 diabetic (T2D) groups.**

| | ND (*n* = 15) | T2D (*n* = 30) | *P*-value |
|---|---|---|---|
| Age (y) | 62 (52–77) | 64 (54–77) | 0.842 |
| Sex M | 8 (53) | 23 (79) | 0.073 |
| COVID duration at inclusion (d) | 14 (5–17) | 11 (7–17) | 0.814 |
| Risk factors for severe COVID-19 | | | |
| BMI (kg/m²) | 22 (21–22) | 28 (24–32) | 0.078 |
| Hypertension (%) | 2 (13) | 8 (62) | **0.003** |
| HbA1c (%) | 5.8 (5.4–6) | 7.8 (7.3–10.3) | **< 0.0001** |
| Severe COVID-19 indicators | | | |
| Intensive care (%) | 4 (27) | 10 (34) | 0.7384 |
| CRP (mg/l) | 138 (51–211) | 146 (86–248) | 0.725 |
| Minimum lymphocytes (10$^9$/l) | 0.75 (0.6–1.04) | 0.9 (0.67–1.12) | 0.590 |
| D-dimers (µg/l) | 2,095 (1,093–3,133) | 1,740 (755–4,505) | 0.971 |

Median (IQR) and Mann–Whitney for qualitative data; *n* (%) and chi-squared or Fisher's exact test for quantitative data. *P*-values in bold represent statistically significant differences between groups (*p* < 0.05). CRP, C-reactive protein; HbA1c, glycated haemoglobin; ND, non-diabetics; T2D, type 2 diabetes.

To address our initial hypothesis of hyperinflammation and a dysregulated immune response to COVID-19 in patients with T2D, we carried out broad-spectrum immunophenotyping of venous blood by multiparametric flow cytometry. Our approach quantifies 15 major innate and adaptive immune populations and subpopulations through lineage (CD45, CD3, HLA-DR, CD14, CD56 and CD20) and phenotypic markers (CD8, CD4, CD16, CD123, CD11c) in a single assay (Appendix Fig S1A). Upon quantification of major immune populations, we confirmed COVID-19-associated lymphopenia (< 20% CD45$^+$ leucocytes) in both ND and T2D patients (Fig 1A and Appendix Fig S1B)(Melzer *et al*, 2015). Unexpectedly, we found a 1.3-fold decrease in CD14$^+$ monocyte frequency in COVID-19 patients with T2D relative to ND patients (Fig 1A and Appendix Table S2). Of note, decreased monocyte frequency was not observed in T2D patients without COVID-19 (Appendix Fig S1B and Table S3). We did not find differences in lymphocyte, natural killer (NK), B cell, granulocyte nor DC frequency between T2D and ND COVID-19 patients. Clinical laboratory full blood count (FBC) records corroborated these findings, with lymphopenia (< 1.5 × 10$^9$/l or < 20% leucocyte count) in ND and T2D COVID-19 patients and a 1.5-fold decrease in monocyte counts in T2D relative to ND COVID-19 patients (Appendix Table S1).

We next quantified monocyte subtypes, to establish whether a specific subtype is affected in T2D patients with COVID-19. Amongst CD14$^+$ monocytes, we found that a 1.4-fold decrease in classical monocyte (CD14$^{Hi}$, CD16$^-$) frequency specifically accounted for the decrease in total monocytes in T2D relative to ND COVID-19 patients (Fig 1B). Of note, a similar trend of decreased (1.5-fold; *P* = 0.07) classical monocyte frequency was also observed in T2D COVID-19 patients when compared to T2D patients without COVID-

19 (Appendix Fig S1C). We did not find any differences in the frequencies of lymphocytic, NK or DC subpopulations between ND and T2D COVID-19 patients, whilst COVID-19-associated lymphopenia was represented across all lymphocyte subtypes in ND and T2D patients (Appendix Fig S1C).

Following these findings, we confirmed with principal component analysis (PCA) including variant immune population frequencies (lymphocyte and monocyte subpopulations) that three distinct clusters of patients emerged corresponding to ND COVID-19, T2D COVID-19 and T2D non-COVID-19 patients (Fig 1C). When the PCA was restricted to monocytic subpopulation frequencies, ND COVID-19 and T2D COVID-19 patients also show robust separation (Appendix Fig S1D). With multivariate ANOVA (MANOVA), integrating previously reported severe COVID-19 risk factors (gender, age, BMI, hypertension and diabetic status), we confirmed that lymphocyte and monocyte subpopulation variance is independently associated with patients with COVID-19 ($P < 0.01$) and T2D ($P < 0.05$; Appendix Table S4).

### Morphologically altered monocytes in type 2 diabetic COVID-19 patients are associated with an aberrant inflammatory response and increased disease severity

To further characterise the monocyte pool, we quantified expression of monocyte activation markers CD16 and CD14, and of the functional pan-antigen-presenting cell (APC) marker HLA-DR. We observed no difference in expression of CD16 nor HLA-DR between

ND and T2D COVID-19 patients. However, T2D COVID-19 patients had 1.5-fold decreased expression of CD14 in their monocytes (Fig 2A and Appendix Fig S2A). These data indicate that monocytes, between both groups of patients, have retained a similar antigen presentation capacity (HLA-DR) and have transitioned away from classical activation (CD16) at the same rate. Monocytes from patients with COVID-19 and T2D have a pronounced loss of CD14, indicating an increased rate of commitment to non-classical activation.

Importantly, we observed a difference in monocyte size, as indicated by forward scatter (FSC) statistics, between T2D and ND patients with COVID-19. Monocyte size was significantly higher in T2D relative to ND patients with COVID-19 (1.4-fold increase; Fig 2A). This unexpected increase in cell size was specific to CD14$^+$ monocytes, as we next compared FSC statistics in CD14 and CD3 bifurcate gates and found no differences between T2D and ND COVID-19 patients in CD14$^-$, CD3$^+$ nor CD3$^-$ cells (Appendix Fig S2B). We next quantified the frequencies of the two distinct populations of FSC-Lo (conventional) and FSC-Hi (large) monocytes in our cohort and corroborated a 1.8-fold higher frequency of FSC-Hi monocytes in T2D relative to ND COVID-19 patients (Fig 2B and Appendix Fig S2C).

To characterise this atypical population of FSC-Hi monocytes, we quantified expression of other known lineage markers (CD20, CD56, CD3) independently of patient status. We found no differences in the expression of non-monocyte markers (CD20, CD56, CD3) between FSC-Lo and FSC-Hi monocytes (Appendix Fig S2D).

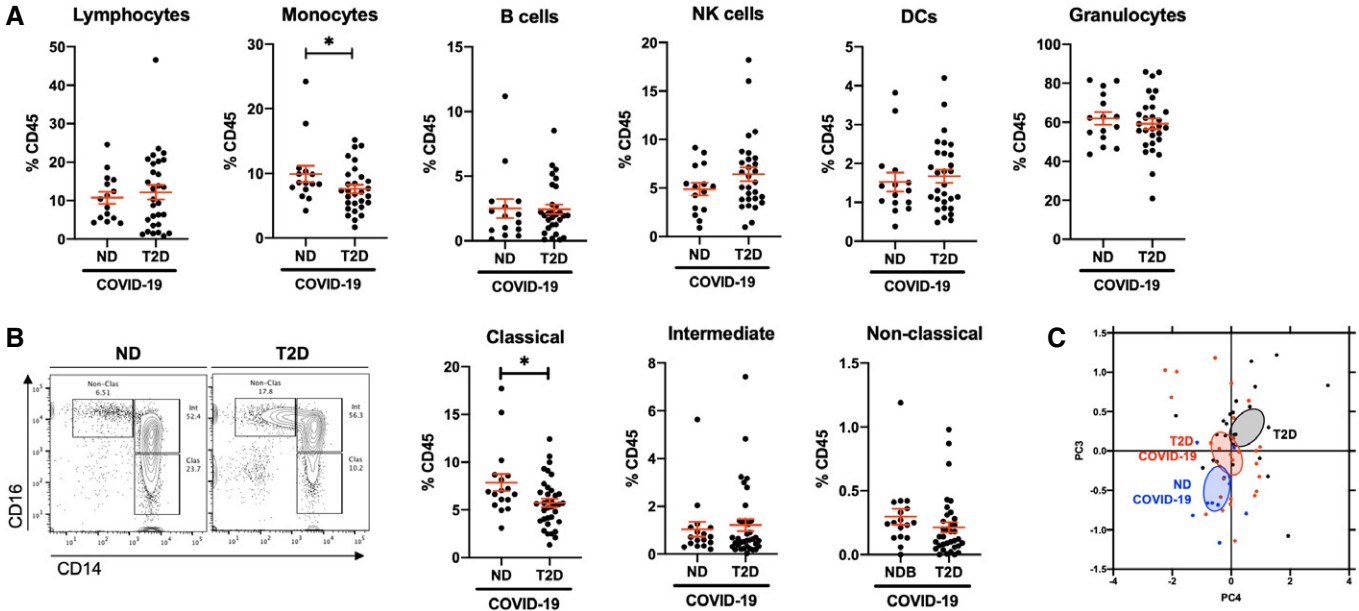

**Figure 1. Type 2 diabetes is associated with decreased monocyte frequency and phenotypic alterations in COVID-19 patients.**

A Flow cytometric quantification of lymphocytes, monocytes, B cells, natural killer (NK) cells, dendritic cells (DCs) and granulocytes in peripheral venous blood samples from non-diabetic (ND) and type 2 diabetic (T2D) patients with COVID-19.

B Quantification of monocyte subpopulation phenotypes as classical (CD14$^{Hi}$ CD16$^-$), intermediate (CD14$^{Hi}$ CD16$^+$) or non-classical (CD14$^{Lo}$ CD16$^+$).

C Principal component analysis of lymphocyte and monocyte population frequencies in ND and T2D COVID-19 patients and in T2D patients without COVID-19.

Data information: Data are presented as mean ± SEM. Differences between groups were evaluated with unpaired *t*-test. *$p < 0.05$. See also Appendix Fig S1 and Tables S1–S3. Sample size and exact *P*-value in Appendix Table S5.

Source data are available online for this figure.

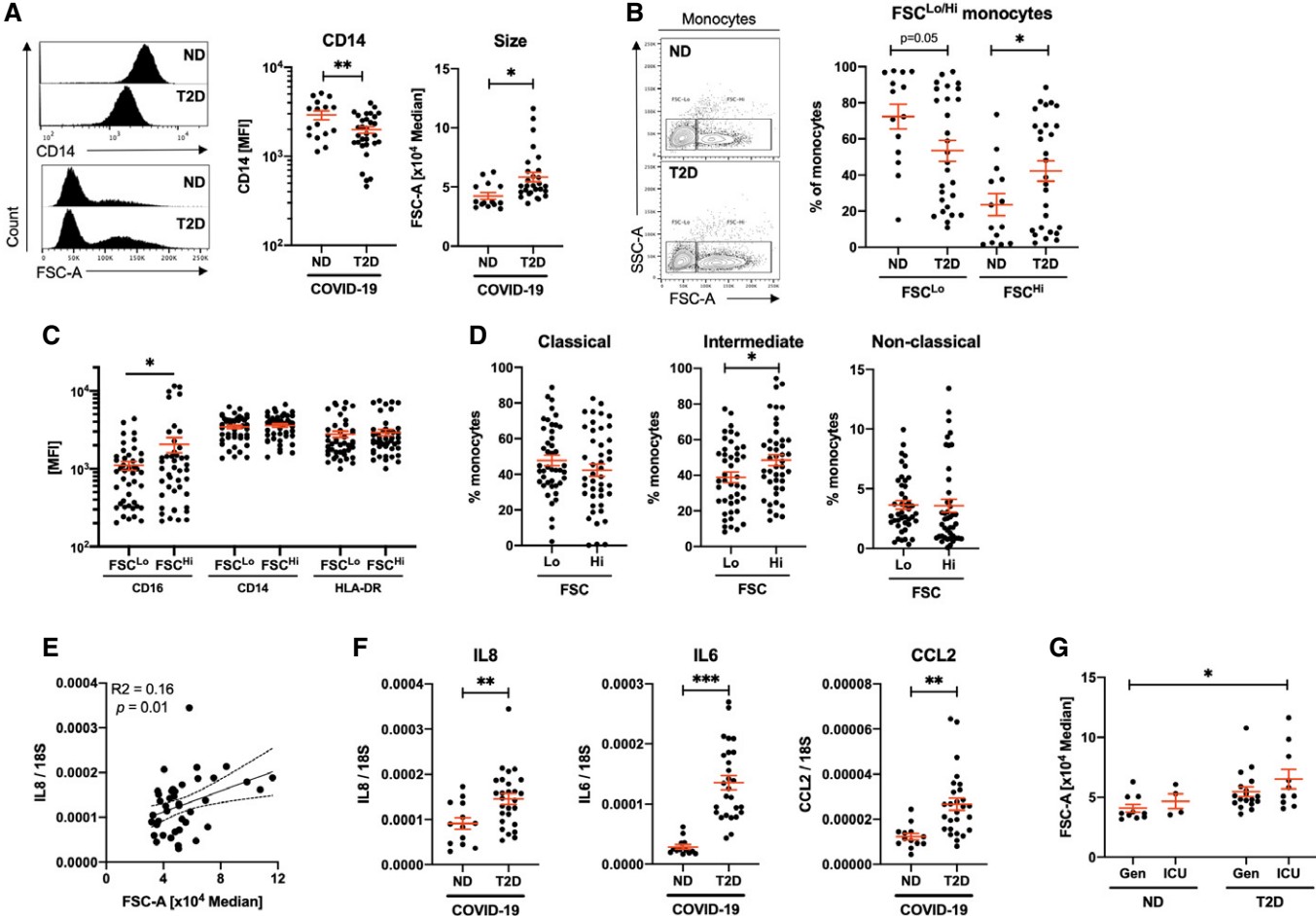

**Figure 2.  Morphologically altered monocytes in type 2 diabetic COVID-19 patients are associated with an aberrant inflammatory response and increased disease severity.**

A   Quantification of CD14 expression and size (FSC) in monocyte populations from non-diabetic (ND) and type 2 diabetic (T2D) COVID-19 patients.
B   Frequency of conventional (FSC^Lo) and large (FSC^Hi) monocytes in ND and T2D COVID-19 patients.
C   Expression of CD16, CD14 and HLA-DR in FSC^Lo and FSC^Hi monocytes from COVID-19 patients.
D   Proportions of classical, intermediate and non-classical monocytes within FSC^Lo and FSC^Hi monocytes from COVID-19 patients.
E   Correlative analyses of IL8 expression in peripheral blood mononuclear cells (PBMCs) to monocyte FSC.
F   IL8, IL6 and CCL2 mRNA expression in PBMCs from ND and T2D COVID-19 patients.
G   Monocyte size quantified in ND and T2D COVID-19 patients treated in general wards (Gen) or in the intensive care unit (ICU).

Data information: Data are presented as mean ± SEM. Differences between groups were evaluated with unpaired *t*-test (except for (C) and (D) where paired by patient). Analysis of variance (ANOVA) or analysis of covariance (ANCOVA) was used for multiple group comparisons. **p* < 0.05; ***p* < 0.01 and ****p* < 0.001. For correlative analysis, Spearman's test was carried out calculating a 2-tailed *P*-value. See also Appendix Fig S2. Sample size and exact *P*-value in Appendix Table S5.
Source data are available online for this figure.

Monocyte activation and functional marker analyses revealed 1.9-fold increased expression of CD16 in FSC-Hi monocytes relative to FSC-Lo monocytes (Fig 2C). CD16 positivity is associated with intermediate and non-classical monocyte class switch. We next quantified the proportions of each monocyte class within FSC-Hi and FSC-Lo populations and found 1.2-fold higher frequency of CD16[+] intermediate monocytes in the FSC-Hi population relative to FSC-Lo (Fig 2D).

Increased phenotypically switched FSC-Hi monocytes and decreased CD14 expression in the T2D monocyte pool (Fig 1B) indicate that patients with COVID-19 and T2D present a

hyperinflammatory phenotype due to exacerbated loss of classical monocytes. To evaluate the extent of the inflammatory response associated with monocyte size and T2D in COVID-19, we carried out gene expression analyses on PBMCs from the patients included in our cohort. Monocyte size was significantly and positively correlated to expression of the pro-inflammatory marker interleukin (IL)-8, whilst other inflammatory markers (IL6, CCL2) trended towards positive correlation to monocyte size (Fig 2E and Appendix Fig S2E). When COVID-19 patients are stratified based on diabetic status, the expression of multiple inflammatory markers is drastically increased in the presence of T2D (1.6- to 4.8-fold increase;

Fig 2F). Corroborating gene expression, circulating levels of IL-8 were also higher in T2D confirming a more pronounced inflammatory state (Appendix Fig S2H). Circulating concentrations, however, did not correlate to mRNA expression in PBMC (Appendix Fig S2I), indicating that IL-8 in circulation is not restricted to that produced by PBMCs, circulating IL-8 may originate from the site of infection. We also investigated IL1B mRNA expression as a marker of inflammation linked to NLRP3 inflammasome activity. We found that although induced in COVID-19, IL1B expression was not different between ND and T2D patients (Appendix Fig S2F), whereas other inflammatory markers related to the type 1 interferon signature were increased in COVID-19 and further increased with comorbid T2D (Appendix Fig S2F).

Given the above findings, we hypothesised that monocyte size would be of prognostic value at presentation. Thus, we stratified patients based on admission to the intensive care unit (ICU) or care in general wards for non-critical patients (Gen). We found monocyte size to be increased by 1.6-fold in T2D patients admitted to the ICU relative to ND patients that did not require ICU admission (Fig 2G). When comparing ICU to Gen, disregarding diabetic status, we did not detect a statistically significant difference in monocyte size (Appendix Fig S2G).

### IRF5 expression is associated with monocyte activation and morphological adaptation in COVID-19 patients

Given the described aetiology of COVID-19 and the patterns of inflammatory marker expression, we hypothesised that the type 1 interferon response is engaged and likely dysregulated in T2D patients. The transcriptional mediator of this pathway is the interferon regulatory factor (IRF)-5, controlling expression of such genes as IL6 and IFNB1. It is highly expressed in the myeloid compartment, is canonically responsive to viral stimuli (Weiss et al, 2015; Forbester et al, 2020) and is involved in regulating metabolic inflammation (Dalmas et al, 2015; Alzaid et al, 2016).

We quantified mRNA expression of IRF5 and of its target type 1 interferon, IFNB1, in PBMCs from our cohort. We found IRF5 and IFNB1 to be induced in ND COVID-19 (respectively, 7.3- and 1.8-fold increase relative to non-COVID-19 patients with T2D; Appendix Fig S3A) and with an exaggerated increase in T2D COVID-19 patients (respectively, 1.5- and 2.0-fold increase relative to ND COVID-19 patients; Fig 3A). Furthermore, expression of IFNB1 correlates to that of IRF5, indicating likely dependent expression in this sample (Fig 3A). Then, we quantified IRF5 expression in circulating cells by FACS and found its highest levels to be in monocytes and other populations with myeloid compartments (DC, NK; Fig 3B and Appendix Fig S3B). Whilst IRF5 expression was not variant in the total monocyte pool between patient groups (Fig 3C and Appendix Fig S3C), we found its expression to increase throughout monocyte activation (Fig 3D). IRF5 expression is increased by twofold in intermediate monocytes of T2D patients relative to ND patients, with a similar trend in non-classical monocytes (Fig 3E and Appendix Fig S3D).

Correlative analyses to monocyte phenotypic and functional markers revealed a positive correlation between IRF5 and HLA-DR, with no correlation to CD14, CD16 nor FSC (Fig 3F and Appendix Fig S3E). These data indicate that IRF5 does not directly regulate monocyte class switch nor morphological changes; however, a dependent relationship exists between IRF5 and HLA-DR.

To evaluate whether the inflammatory response mediated by IRF5 could infer protection or severity in COVID-19, we stratified our patients based on their admission to the ICU. In the monocyte pool, we observed a 2.1-fold increase in expression in patients admitted to the ICU, significant only amongst ND COVID-19 patients (Fig 3G). Interestingly, in monocyte subpopulations we observed a specific signature associated with ICU admission taking into account patient diabetic status. Classical monocyte expression of IRF5 is specifically increased by 2.1-fold in patients admitted to the ICU, whereas intermediate monocyte expression of IRF5 is only increased in T2D patients admitted to the ICU (respectively, 3.3- and 1.7-fold relative to ND and T2D not requiring ICU admission; Fig 3H).

### Severity of COVID-19 is associated with loss of CD8[+] lymphocytes and loss of classical monocytes in patients with T2D

Despite the majority of ICU admissions from our cohort being patients with T2D, the proportions of ICU admissions within ND patients were comparable (26.7% of ND versus and 34.5% of T2D; Fig 4A). To decipher the immunophenotypic and inflammatory signatures more specifically associated with COVID-19 severity in T2D, we compared mild–moderate and severe–critical cases of COVID-19 in this patient group.

When patients with T2D are stratified based on ICU admission, we observe a significant 1.6-fold decrease in monocyte frequency and a trend to decreased lymphocyte frequency (1.8-fold) in T2D patients requiring ICU admission relative to those treated in general wards (Gen; Fig 4B). The decrease in lymphocyte and monocyte proportions is corroborated by hospital-issued FBCs (Appendix Fig S4A). Quantification of immune subpopulations revealed concomitant decreases in CD8[+] T cells (2.0-fold) and classical monocytes (1.3-fold) in ICU, the latter of which confirms monocytopenia, specific to classical monocytes, as a hallmark of COVID-19 infection in patients with T2D (Fig 4C). A further loss of classical monocytes is indicative of severity in T2D. No other populations in circulation were significantly affected in the ICU group (Appendix Fig S4B and C).

We next applied unsupervised analyses to our cytometry data, consisting of dimensional reduction by the t-SNE algorithm (van der Maaten & Hinton, 2008). Marker mapping was performed by projecting known lineage markers to the t-SNE map and we evaluated variation between patients in this analyses. Visual analysis of t-SNE maps confirmed the monocyte and lymphocyte dynamics associated with T2D and with COVID-19 severity, notably a marked loss of CD14[+] monocytes and CD8[+] lymphocytes in T2D patients with severe COVID-19, versus a specific loss of CD8[+] lymphocytes in ND patients with severe COVID-19 (Fig 4D). When projecting IRF5 expression over the t-SNE maps, we confirm monocyte-specific expression, maintained at a high level despite marked monocytopenia in severe COVID-19 with T2D.

## Discussion

2020 has been marred by the emergence of SARS-CoV-2 affecting over 4.5 million people globally. The resulting coronavirus disease (COVID)-19 has placed unprecedented strain on health services and claimed over 300 thousand lives (Dong et al, 2020). With diabetic patients accounting for approximately 10% of COVID-19 deaths,

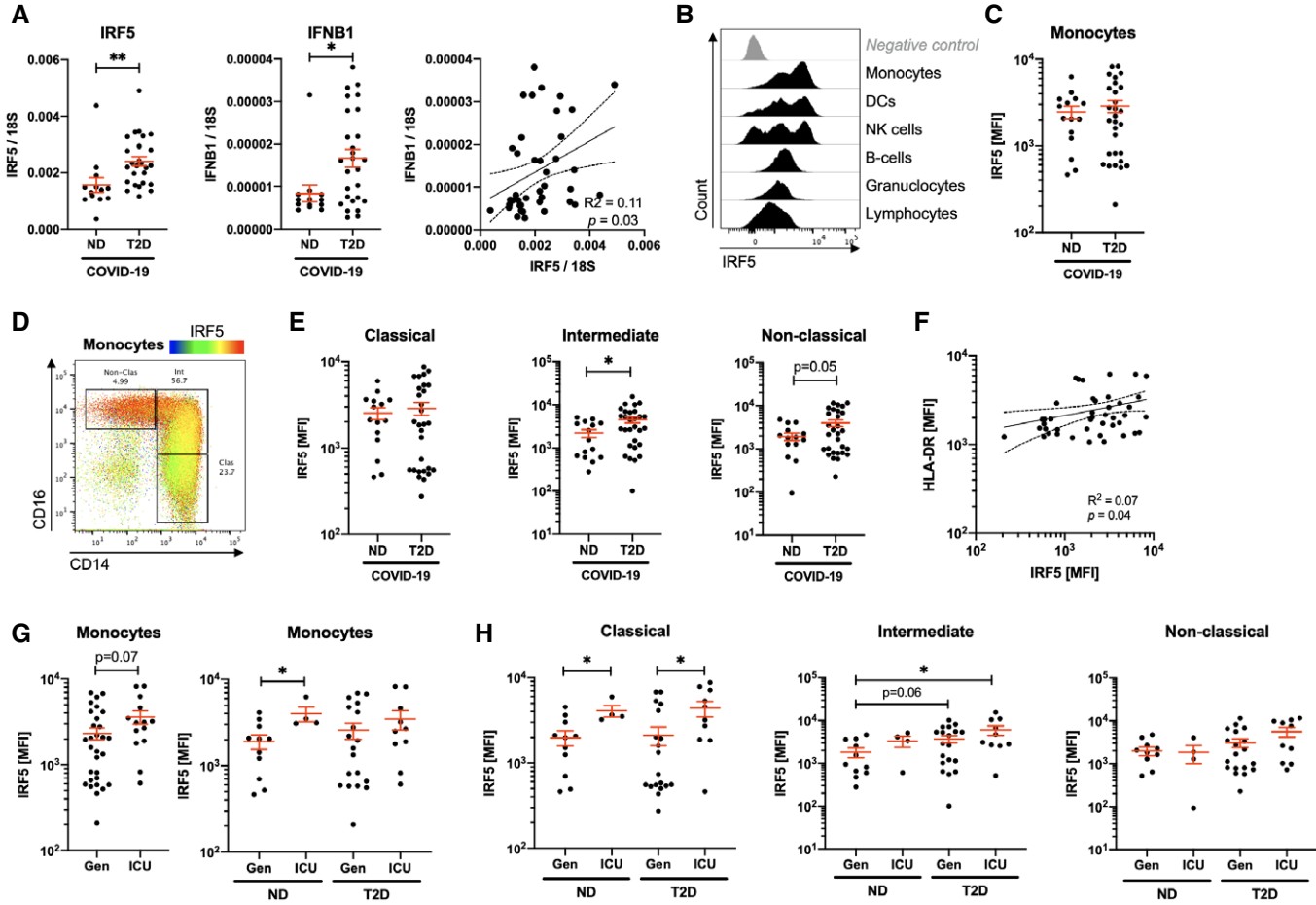

**Figure 3. IRF5 expression is associated with monocyte activation and morphological adaptation in COVID-19 patients.**

A    Quantification of IRF5 and IFNB1 mRNA expression in peripheral blood mononuclear cells of non-diabetic (ND) and type 2 diabetic (T2D) COVID-19 patients.
B    Histograms of IRF5 expression on different populations analysed by flow cytometry on venous blood samples.
C    IRF5 median fluorescence intensity (MFI) in monocytes of ND and T2D patients with COVID-19.
D    IRF5 expression overlaid onto monocyte phenotypic gating.
E    IRF5 expression (MFI) in monocyte subtypes from ND and T2D COVID-19 patients.
F    HLA-DR and IRF5 expression in monocytes from COVID-19 patients.
G    IRF5 expression in monocytes of ND or T2D COVID-19 patients admitted to the intensive care unit (ICU) or treated exclusively in general wards (Gen).
H    IRF5 expression in monocytes of ND or T2D COVID-19 patients admitted to the ICU or treated exclusively in Gen.

Data information: Data are presented as mean ± SEM. Differences between groups were evaluated with unpaired $t$-test. Analysis of variance (ANOVA) was used for multiple group comparisons. *$p < 0.05$ and **$p < 0.01$. For correlative analysis, Spearman's test was carried out calculating a 2-tailed $P$-value. See also Appendix Fig S3. Sample size and exact $P$-value in Appendix Table S5.
Source data are available online for this figure.

urgent mobilisation of diabetologists and diabetes researchers is required to define criteria that confer risk. An example of such valuable work has found that appropriate glycaemic control is a key determinant of survival amongst T2D patients with COVID-19 (Zhu *et al*, 2020). Our current study investigates inflammation and immune cell dynamics as possible mechanisms conferring risk of severe COVID-19 in patients with T2D.

In COVID-19 patients with or without T2D, we carried our broad-spectrum immunophenotyping of 15 leucocyte populations and subpopulations in peripheral circulation, evaluating changes associated with diabetic status and disease severity. We found that T2D patients with COVID-19 are characterised by monocytopenia,

specific to quiescent monocytes. Monocyte loss was accompanied by morphological alterations and a hyperinflammatory expression profile consistent with the type 1 interferon response. These phenomena, alongside a marked loss of CD8[+] T cells in peripheral circulation, characterise severe COVID-19 in pre-existing T2D.

**Lymphopenia, monocytopenia and phenotypic variation in COVID-19 with pre-existing T2D**

In keeping with recent reports, we confirmed lymphopenia as a hallmark of COVID-19 (Tan *et al*, 2020). What we were astounded to observe was the added monocytopenia in patients with T2D. To our

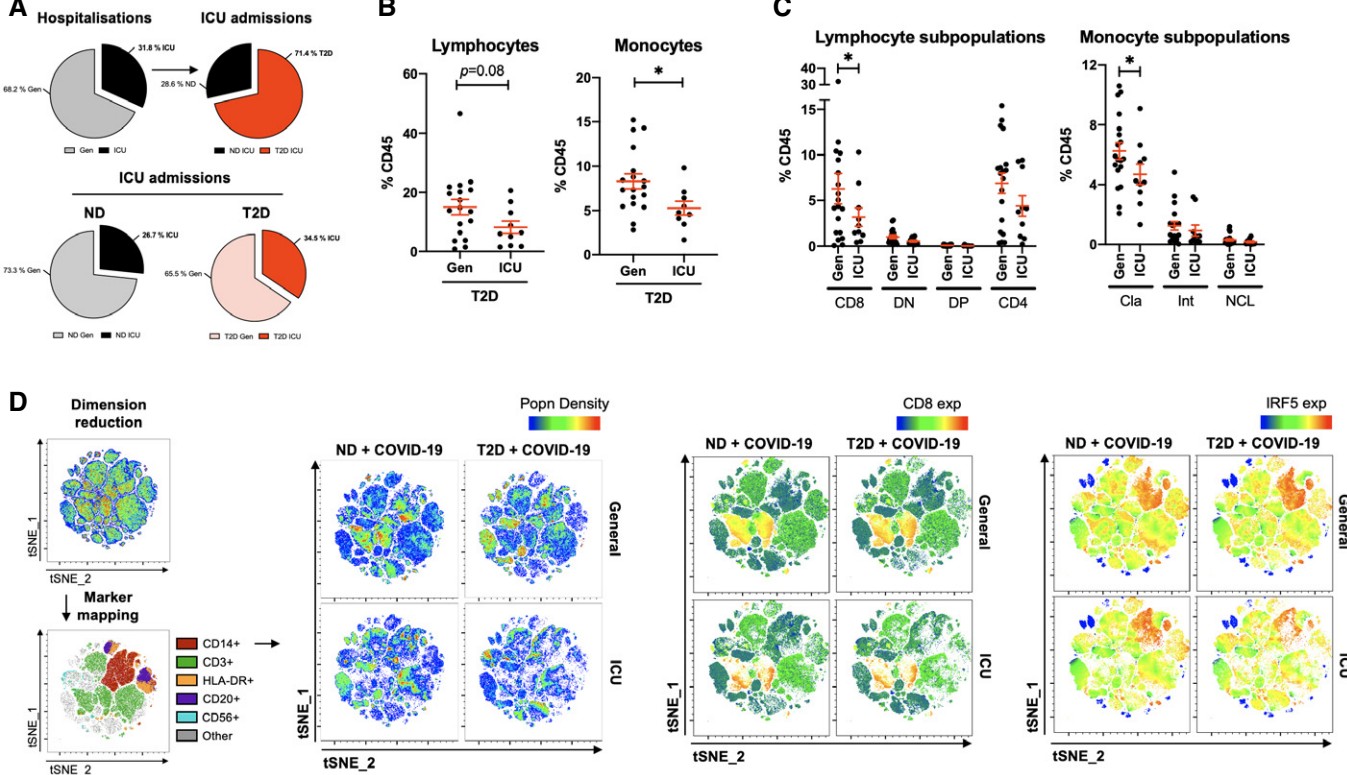

**Figure 4. Severe COVID-19 is associated with loss of CD8+ lymphocytes and loss of classical monocytes in patients with T2D.**

A Pie charts of COVID-19 hospitalisation that were exclusively treated in general wards (Gen) or that required intensive care (ICU). Proportions of patients admitted to the ICU that were non-diabetic (ND) or with type 2 diabetes (T2D).

B Proportions of lymphocytes and monocytes in peripheral venous blood from T2D patients with COVID-19 either treated in Gen or requiring ICU admission.

C Phenotypic analysis of lymphocyte and monocyte subpopulations in peripheral venous blood from T2D COVID-19 patients treated in Gen or requiring ICU treatment.

D t-SNE mapping of all cytometric acquired data with projections of population density, CD8 or IRF5 expression. Maps are of representative profiles from each group.

Data information: Data are presented as mean ± SEM. Differences between groups were evaluated with unpaired *t*-test or non-parametric Mann–Whitney *U*-test.
*$p < 0.05$. See also Appendix Fig S4. Sample size and exact *P*-value in Appendix Table S5.
Source data are available online for this figure.

knowledge, ours is the first report of monocytopenia specific to T2D patients suffering a coronavirus infection, and such is likely to be a particularity of COVID-19. Monocytopenia has been previously associated with acute infection (Jardine *et al*, 2019), whilst, in our current study, duration of COVID-19 was not different between groups and thus monocytopenia is likely specifically related to patient diabetic status. Of note, monocyte loss detected by flow cytometry is corroborated by clinical FBC and thus could serve as an additional COVID-19 screening tool for diabetology practice, with lymphopenia as an established indicator of severity in the general population.

Beyond the quantitative analyses, we identified an exaggerated phenotypic class switch of monocyte subpopulations in COVID-19 patients with T2D. As monocytes undergo activation, they transition from their classical phenotype to intermediate and non-classical phenotypes gaining expression of CD16 and gradually lowering expression of CD14 (Hijdra *et al*, 2013). We demonstrate an increased rate of transition from the classical phenotype, towards CD16+ intermediate monocytes, and an accelerated loss of CD14 towards the non-classical phenotype in T2D. Both chronic

inflammation and viral infection (in particular, influenza viruses) are associated with switching monocyte class (Hoeve *et al*, 2012; Friedrich *et al*, 2019). Importantly, monocyte expression of CD16 increases potency of inducing CD4+ T-cell polarisation compared to quiescent (classical) monocytes that are CD16− (Sanchez-Torres *et al*, 2001). This could in part explain the hyperinflammatory response in patients with T2D that have accelerated monocyte activation and a marked loss of quiescent monocytes.

A specific loss of classical monocytes may indeed be due to increased apoptosis when confronted with a viral challenge, as has been reported in *in vitro* modelling (Hoeve *et al*, 2012). Alternatively, monocytes in T2D patients may be primed to transition towards their activated damage-seeking state more rapidly than in ND patients. Of note, the accumulation of intermediate and non-classical monocytes is associated with increased rate of infection and a largely maladaptive response (Hoeve *et al*, 2012; Coates *et al*, 2018; Vangeti *et al*, 2019). Moreover, non-classical monocytes have recently been reported to important drivers of pulmonary hypertension in T2D patients (Yu *et al*, 2020). Taken together, these data are coherent in explaining, in part,

increased risk of severe disease or mortality in COVID-19 patients with T2D.

## Morphological alterations and hyperinflammation in COVID-19 patients with T2D

A recent pre-print reported that monocyte morphological alterations are associated with COVID-19 (Zhang et al, 2020). These alterations consisted of increased monocyte size (FSC^Hi monocytes) in a subset of COVID-19 patients. Interestingly, altered monocyte size has also been reported in infections of influenza virus (Hoeve et al, 2012). A major novelty of our investigation is the characterisation of these morphologically altered monocytes in T2D. We corroborate the recent pre-print report of FSC^Hi monocytes in COVID-19, and we demonstrate an increased frequency of FSC^Hi monocytes in T2D. Moreover, we demonstrate that monocyte size is associated with a hyperinflammatory gene expression profile in PBMCs and with admission to ICU of patients with T2D. Surprisingly, little is known with regard to the regulation of monocyte morphology; further basic research is required to decipher the natural history and functional implications of such phenomena.

## Exacerbated IRF5-linked hyperinflammation in T2D patients with COVID-19

With prior knowledge of pathways engaged in both viral and diabetic contexts, we targeted a factor that controls transcription of type 1 interferons, IRF5. Previous studies have well-characterised the pro-inflammatory function of IRF5 in tissue macrophages, the progeny of circulating monocytes (Dalmas et al, 2015; Weiss et al, 2015). In the current study, we demonstrate that IRF5 expression in circulating cells is induced in COVID-19 and is increased in activated monocytes of patients with T2D relative to ND patients. Indeed, expression is correlated to that of inflammatory markers. A specific increase in activated subtypes of monocytes is in agreement with previous reports that IFR5 drives monocyte damage-seeking behaviour and differentiation to inflammatory macrophages (Yang et al, 2012; preprint: Corbin et al, 2019). Whilst IRF5 expression did not correlate to monocyte activation markers, we did observe a positive correlation between IRF5 and HLA-DR. A positive correlation indicates a possible dependent relationship where HLA-DR may form part of a mechanism feeding back to increase IRF5 expression; alternatively, IRF5 may impact monocyte antigen presentation capacity or other HLA-DR-associated functions. Moreover, increased expression of IRF5 in different compartments of the monocyte pool also marked those patients that required intensive care.

The molecular mechanisms linking hyperglycaemia and cellular glucose metabolism directly to an IRF5-dependent cytokine storm have recently been described in the case of influenza A virus (IAV) infection, of which some mechanisms may be shared with COVID-19 (Wang et al, 2020a). Indeed, increased glucose consumption is characteristic of inflammatory effector function of macrophages, where glucose shuttling to the hexosamine biosynthesis pathway provides a substrate for O-GlcNAcylation of IRF5 on serine-430 and its subsequent K63-linked ubiquitination. These post-translational modifications allow the downstream processing of IRF5 and the engagement of its pro-inflammatory transcriptional activities. IRF5 O-GlcNAcylation in human PBMC and subsequently increased IL8

and IL6 levels in circulation were associated with increased blood glucose in IAV-infected patients (Wang et al, 2020a). The IRF5 expression and cytokine profile reported in IAV infection are similar to what we observe in PBMC from SARS-CoV-2-infected patients, comforting our hypothesis that similar mechanisms are at play. Importantly, the K63-linked ubiquitination of IRF5, required for its nuclear translocation, is an indispensable mechanism in macrophages that mediate metabolic inflammation and loss of glycaemic homeostasis in T2D (independently of viral infection) (Kim et al, 2017). A recent report has also dissociated IRF5-mediated cytokine production from an inhibitory effect on viral replication in IAV. These studies, and a recent pre-print implicating impaired type 1 interferon signalling in severity of COVID-19 cases, strongly support a key role for IRF5 and the type 1 interferon response in increased severity in T2D (Forbester et al, 2020; preprint: Hadjadj et al, 2020).

Taken together, our data and previous reports indicate that basal levels of IRF5, preceding-SARS-CoV-2 infection, are dysregulated in T2D patients; our supposition is comforted by a recent opinion article describing the hypothetical molecular mechanisms (Laviada-Molina et al, 2020). Monocytopenia and rapid class switch of monocytes in T2D with COVID-19 may be the result of an exuberant viral response from an immune system primed on an inflammatory background. IRF5-linked hyperinflammation will induce eager damage-seeking behaviour, antigen presentation and cytokine release, without affecting viral replication, thus contributing to the cytokine storm syndrome that characterises severe COVID-19 (Gruber, 2020).

## CD8[+] T-cell loss and monocytopenia are hallmarks of severe COVID-19 in T2D

In our cohort, as in others, not all COVID-19 patients with T2D require intensive care, approximately 10% more T2D patients required admission to ICU than ND patients. The immunophenotype of patients with T2D that required ICU admission was marked by an exacerbated monocytopenia and lymphopenia; in the case of lymphocyte phenotypes, we were surprised to find that cytopenia was specific to CD8[+] T cells. A recent correspondence also highlighted a loss of CD8[+] cytotoxic lymphocytes (CTLs) in patients with COVID-19, both severe and mild forms (Zheng et al, 2020). The CTL population is known to play a crucial role in recognising and killing virally infected cells (Koutsakos et al, 2019). This study by Koutsakos et al demonstrates that CTLs are important in conferring cross-reactivity across a number of viral strains, in effect a key target population that holds important clues for neutralising vaccine development. It is thus not surprising that those patients with under-represented CD8[+] T cells develop a more severe form of disease. A higher frequency of CD8[+] T cells and their efficient interactions with APCs, such as monocytes, may be a key process in mitigating severe disease.

Finally, unsupervised clustering of immune populations based on expression data also mapped a marked loss CD8[+] T cells to ICU admission of both T2D and ND patients with COVID-19, as well as a loss of monocytes in T2D patients with severe disease. Importantly, when IRF5 is projected onto these clusters, we observe a sustained high level of expression specific to monocytes, even amongst populations that decrease in frequency. These data indicate that in T2D, exuberant inflammation, likely mediated by IRF5 in monocytes,

may be responsible for increased mobilisation of these cells. Their increased mobilisation decreases their abundance in peripheral circulation or at sites of interaction with lymphocytic populations. A mismatched ratio of antigen-presenting monocytes to CD8$^+$ T cells is a tempting mechanism for the descent from mild–moderate COVID-19 to severe–critical COVID-19. Once patients are identified as at-risk at admission, well-timed and well-targeted immunosuppressive intervention may prevent the uncontrolled and persistent inflammatory response in severe or mortal COVID-19.

### Limitations

A main limitation of our study is the sample size and diversity; inclusion of veritable mild cases that did not require hospitalisation would help support and extend our conclusions with regard to the factors that increase COVID-19 severity. Increasing sample size and diversity within the T2D group would also allow the further stratification of patients based on severity of diabetes, in terms of glycaemic control or the existence of complications or comorbidities. Although the current study hypothesised that T2D itself was a risk factor through increasing systemic inflammation, thus, we aimed and achieved the recruitment of groups comparable in terms of age, BMI, gender and disease duration (as well as clinical factors that indicate severity such as CRP and D-dimers). Lastly, the *cytokine storm* syndrome was not documented in this cohort; however, several publications have very concretely established its presence in COVID-19 cases. Given the time-sensitive nature of this public health problem, we privileged novel analyses to directly address the hypothesis.

## Materials and Methods

### Human populations

All consecutive patients hospitalised in April 2019 for COVID-19 infection in the dedicated University Centre of Diabetes and its Complications, Lariboisière Hospital, Paris, France, were prospectively included in the study. Clinical and anthropometric data are summarised in Table 1. COVID-19 diagnosis was based on positive oropharyngeal swab RT–PCR. Exclusion criteria were type 1 diabetes, immunotherapy for previous transplantation and treatment of COVID-19 by corticotherapy or immunotherapy before inclusion. For the analysis, patients were divided into two groups: with or without T2D. Patients with T2D are patients already known to have it before hospitalisation or patients for which diabetes has been discovered during hospitalisation (HbA1c > 6.5% or two fasted glycaemia above 7 mmol/l). The study was conducted in accordance with the Helsinki Declaration and was registered in a public trial registry (ClinicalTrials.gov; NCT02671864). This study was approved by local institutions and ethical committees, and the Ethics Committee of CPP Ile-de-France granted approval for all individuals (Ile de France V number 15070). Non-COVID-19 T2D patients were included from the same clinical service prior to the COVID-19 pandemic as part of an ongoing observational study (NCT02671864); patients were matched to the T2D COVID-19 cohort in terms of age, gender, BMI, hypertension and HbA1c. All patients provided informed consent indicating that they understood the

nature of their participation in the study (NCT02671864). The principal investigator of this clinical trial is Prof. Gautier Jean-François: jean-francois.gautier@aphp.fr.

### Data collection

Sampling, clinical data (age, gender, BMI, diabetes duration, history of HTA, COVID duration since 1s symptoms) and biological data (HbA1c, CRP, lymphopenia, D-dimer) were collected at inclusion in a standardised manner; events like introduction of COVID-19 specific treatments, transfer in intensive care unit (ICU) and death were collected at the end of the hospitalisation. Minimum lymphocyte count and maximal CRP were collected at the end of the hospitalisation.

### Immunophenotyping by flow cytometry

Blood cells were obtained from 1 ml of venous blood, after red blood cell lysis, and resuspended into FACS buffer as previously described (Julla *et al*, 2019). After 10-min incubation with an Fc blocker (120-000-422; Miltenyi), cells were stained for surface markers with the appropriate antibodies and a Live/Dead viability dye (L34957; Thermo Fisher Scientific) according to manufacturer's protocol. The following antibodies were used: anti-HLA-DR (AC122) and anti-CD8 (BW135/80) from Miltenyi; anti-CD14 (MΦP9), anti-CD3 (UCHT1) and anti-CD123 (7G3) from BD Biosciences; anti-CD16 (3G8), anti-CD56 (HCD56), anti-CD20 (2H7) and anti-CD11c (3.9) from BioLegend; and anti-CD4 (S3.5) and anti-CD45 (HI20) from Thermo Fisher Scientific. After washing, cells were fixed and stained using the Foxp3-staining kit (00-5523-00; Thermo Fisher Scientific) according to the manufacturer's protocol and using the anti-IRF5 (ab21689; Abcam) antibody for 1 h at 4°C in the dark, followed by the donkey-anti-rabbit-PE (12-4739-81; Thermo Fisher Scientific) secondary antibody for 20 min at 4°C in the dark. Antibodies were diluted in the following ratios: 1:200 for anti-HLA-DR, anti-CD14, anti-CD45, anti-CD3, anti-CD16, anti-CD20, anti-CD4, anti-CD8 and anti-IRF5; 1:100 for anti-CD56 and anti-CD123; and 1:50 for anti-CD11c and for the secondary PE antibody. Viability dye was diluted at 1:1,000 following the manufacturer's protocol. Acquisition was performed on a LSRFortessa flow cytometer (BD Biosciences) and analysed with FlowJo software (Tree Star). Immune lineage markers and antibodies were chosen based on the Immunological Genome Consortium guidelines, and gating strategies have been previously published (Heng & Painter, 2008; Autissier *et al*, 2010; Melzer *et al*, 2015).

### RT–qPCR analysis

RNA was extracted from blood cells (PBMCs) using the RNeasy RNA Mini Kit (Qiagen). Complementary DNAs were synthesised using SuperScript Kit (Promega). RT–qPCR was performed using the QuantStudio 3 Real-Time PCR Systems (Thermo Fisher Scientific). 18S was used for normalisation to quantify relative mRNA expression levels. Relative changes in mRNA expression were calculated using the comparative cycle method ($2^{-\Delta\Delta C_t}$). Primer sequences were designed using Primer3 (Koressaar & Remm, 2007; Untergasser *et al*, 2012) (http://bioinfo.ut.ee/primer3-0.4.0/) used: IL8 (F: AG ACAGCAGAGCACACAAGC; and R: ATGGTTCCTTCCGGTGGT), CCL2

(F: TTCTGTGCCTGCTGCTCAT; and R: GGGGCATTGATTGCATCT),
IRF5 (F: GATGGGGACAACACCATCTT; and R: GGCTTTTGTTAAGG
GCACAG), IL6 (F: GCCCAGCTATGAACTCCTTCT; and R: GAAGGC
AGCAGGCAACAC), IFNB1 (F: GGAAAGAGGAGAGTGACAGAAAA;
and R: TTGGATGCTCTGGTCATCTTTA) and 18S (F: TTCGAACGTC
TGCCCTATCAA; and R: ATGGTAGGCACGGCGACTA).

## Statistical analyses

Data are expressed as mean or median $\pm$ SD or SEM as indicated in
the text, tables and figures. For quantitative data, differences between
groups were evaluated with non-parametric Mann–Whitney $U$-test,
and categorical and binary variables were tested by the two-tailed
Pearson chi-square test or Fisher exact test if more than 20% of the
cells in the frequency tables had an expected frequency below 5.
Analysis of variance (ANOVA) or analysis of covariance (ANCOVA)
was used for group comparisons. Repeated measures multivariate
analyses of variance (MANOVA) was employed to compare the vari-
ance of monocyte populations taking into consideration gender, BMI,
hypertension and diabetic status. Statistical analyses were carried out
using JMP (SAS Institute Inc, Cary, NC), XLSTAT 2014 (Addinsoft,
Brooklyn, NY), GraphPad Prism (GraphPad), SPSS Statistics (SPSS
corporation) and R Software 3.6.0 (http://www.r-project.org). Statis-
tical significance was set at $P < 0.05$. Principal component analysis
(PCA) was performed from total lymphocyte and monocyte subpopu-
lation FACS quantification with FactoMineR R package (https://doi.
org/10.18637/jss.v025.i01), and factoextra package (factoextra.bib)
was used to construct graphics. Contribution of each individual was
analysed to identify outliers biasing the PCA, leading to exclusion of
4 individuals for a technical bias or hospitalisation in the ICU prior to
COVID-19 infection. Given the pressing circumstances during COVID-
19 pandemic, a power study was not carried out prior to recruitment;
sample size was judged sufficient when differences of immunopheno-
typic were observed. For detailed sample size and exact $P$-value per
figure panel, see Appendix Table S5.

# Data and software availability

The computer code and software tools used in this study are
available:

R Software 3.6.0 (http://www.r-project.org). R Core Team (2013).
FactoMineR R package (https://doi.org/10.18637/jss.v025.i01)
Factoextra: Extract and Visualize the Results of Multivariate Data
Analyses 1.0.7. https://CRAN.R-project.org/package = factoextra
t-Distributed Stochastic Neighbor Embedding (t-SNE). https://jmlr.
org/papers/v15/vandermaaten14a.html
Raw and source data can be accessed through the Zenodo online
data repository https://doi.org/10.5281/zenodo.3987326 [Open
access as of publication date].

**Expanded View** for this article is available online.

## Acknowledgements

F.A. was supported by grants from the French National Agency of Research
(ANR) for ANR MitoFLAME (ANR-19-CE14-0005) and the European Foundation
for the Study of Diabetes (EFSD)/Lilly grant (Characterisation of monocyte

## The paper explained

### Problem

COVID-19 severity is associated with progressive lymphocytopenia and
systemic inflammation. Type 2 diabetes (T2D) is characterised by
chronic low-grade inflammation and is a risk factor for severe COVID-
19. Is COVID-19 severity in T2D associated with distinct inflammatory
and immunophenotypic variations?

### Results

COVID-19 and its severity with pre-existing T2D are associated with
monocytopenia and altered morphology. Interferon-driven inflamma-
tion and monocyte phenotypes are also perturbed in COVID-19 with
T2D. Severe COVID-19 is associated with lymphocytopenia of CD8[+]
cells in the general population, whilst monocytopenia of classical
monocytes is a specific to patients with T2D.

### Impact

Monocyte frequency and morphology are fast and reliable metrics to
assess COVID-19 severity in patients with T2D. The inflammatory pro-
file we observed supports the therapeutic application of anti-inflam-
matory drugs.

metabolism and bioenergetic responses in type-2 diabetes and risk of cardio-
vascular disease). N.V. was supported by grants from the French National
Agency of Research (GLUTADIAB and ANGIOSAFE), the European Foundation
for Diabetes (EFSD) and the European Union H2020 Framework (ERC-
EpiFAT 725790). The human study was performed at the Clinical Investigation
Centre (Groupe Hospitalier Saint-Louis/Lariboisière, Paris) and was supported
by Assistance Publique des Hôpitaux de Paris (ANR-DGOS Project AngioSafe
T2D; J.-F.G., principal investigator) and ASSERADT (a non-profit patient associa-
tion). E.D was supported by grants from the French and European Foundation
for Diabetes (SFD and EFSD). The authors would like to thank Hélène Fohrer-
Ting of the Centre for Cytometry, Histology and Cellular Imaging (CHIC) core
facility of the Cordeliers Research Centre (CRC), for her guidance, patience and
support. The authors thank the Unité de Recherche Clinique of Lariboisière/
Fernand Widal (Pr Eric Vicaut & Dr Véronique Jouis); the Délégation à la
Recherche Clinique de Paris—Ile de France; and the Direction de la Recherche
Santé de Marseille for their administrative support. The authors also thank
Nassima Haddadi, Djamila Bellili and Hanane Mersel for their technical
support, and nurses and patients who accepted to be involved in these studies.
The authors also thank Jagadeesh Bayry and Anupama Karnam for their gener-
ous gift of the IL-8 ELISA Kit.

## Author contributions

FA, J-BJ, MD, ED, NV and J-FG designed research studies, analysed data and
wrote the manuscript. J-BJ, TV-T, J-PR, J-FG, PM and BG carried out patient
inclusion. FA, J-BJ, CP and NV conducted experiments, and acquired and anal-
ysed data. MD and GV carried out data and statistical analyses. BG, SG, LP, RR
and J-PR helped with experimental design and scientific discussion.

## Conflict of interest

The authors declare that they have no conflict of interest.

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
