## [Review Process File · EMBO Molecular Medicine]

Monocytopenia, monocyte morphological anomalies and hyperinflammation characterise severe COVID-19 in type 2 diabetes

Fawaz Alzaid, Jean-Baptiste Julla, Marc Diedisheim, Charline Potier, Louis Potier, Gilberto Velho, Benedicte Gaborit, Philippe Manivet, Stephane Germain, Tiphaine Vidal-Trecau, Ronan Roussel, Jean-Pierre Riveline, Elise Dalmas, Nicolas Venteclef and Jean-François Gautier

DOI: [10.15252/emmm.202013038](https://doi.org/10.15252/emmm.202013038)

Corresponding authors: Fawaz Alzaid (fawaz.alzaid@gmail.com) , Jean-François Gautier (jean-francois.gautier@aphp.fr)

Review Timeline:

Submission Date:	30th Jun 20
Editorial Decision:	21st Jul 20
Revision Received:	3rd Aug 20
Editorial Decision:	14th Aug 20
Revision Received:	17th Aug 20
Accepted:	19th Aug 20

Editor: Celine Carret

Transaction Report:

21st Jul 2020

Dear Dr. Alzaid,

Thank you for the submission of your manuscript to EMBO Molecular Medicine. We have now heard back from the three referees whom we asked to evaluate your manuscript. As you will see from the reports below, the referees find the topic of your study of interest. Still, they raised concerns that should be directly addressed in a revised article. Of relevance, we would like to insist on the evaluation of more clinical parameters, re-working the introduction and discussion to put them in a broader context and finally adding details and explanations as suggested.

We would therefore welcome the submission of a revised version within three to six months (or as soon as possible given the topic) for further consideration and would like to encourage you to address all the criticisms raised as suggested to improve conclusiveness and clarity. Please note that EMBO Molecular Medicine strongly supports a single round of revision and that, as acceptance or rejection of the manuscript will depend on another round of review, your responses should be as complete as possible.

I look forward to seeing a revised form of your manuscript as soon as possible.

Yours sincerely,

Celine Carret

Celine Carret, PhD
Senior Editor
EMBO Molecular Medicine

***** Reviewer's comments *****

Referee #1 (Comments on Novelty/Model System for Author):

The manuscript addresses highly relevant question about the changes in the immune cell populations and level of inflammatory parameters in blood circulation that are caused by COVID-19 in the presence and absence of type 2 diabetes. The study uses representative cohort of patients. The authors used state-of-the art biomarkers for monocytes in order to characterise major populations. The novelty of the study is that diabetic patients respond to COVID-19 by decreasing of frequency of monocytes and phenotypic alterations of monocyte subsets. These changes are paralleled by the enhanced inflammatory response. Highly interesting is specific upregulation of IL8 and IRF5 expression in PBMCs of diabetic patients in response to COVID-19. The study provides convincing data showing that parameters of circulating monocytes are indicative for the hyper-inflammation that is found in diabetic patients infected by COVID-19 and that correlates with the severity of virus-induced pathology.

Referee #1 (Remarks for Author):

Major comments

- 1.The data about the changes monocytes amounts and amounts of total PBMC per ml of blood in nondiabetic versus diabetic patients in response to COVID-19 have to be provided.
- 2.IL8 concentrations have to be measured in the serum of patients. Do secreted levels of IL-8 directly correlate with gene expression in PBMC? Are circulating levels of IL8 biologically significant?
- 3.Which medications have been used for the diabetic patients? Which subgroups can be defined based on the treatment? How anti-diabetic medications affect monocytes subsets and morphology?
- 4.Which vascular complications have been found in diabetic patients? Do monocytes changes correlate with vascular complications depending on COVID-19 infection?

Minor comments

1. It is recommended to make more straight forward formulations in the abstract listing what are the major differences in monocytes phenotype identified in non-diabetic versus diabetic patients in response to COVID-19 infection
2. Fold change for all measured parameters has to be indicated in the text of Results. The biological significance of the fold change has to be commented.

Referee #2 (Remarks for Author):

This interesting paper evaluates differences in 15 leukocyte populations between COVID-19 patients with and without T2D and assesses if these differences are associated with the severity of disease in each population. The central finding of this paper is the identification of monocytopenia (linked to a loss of classical monocytes) in T2D patients with COVID-19 and an association between this dysfunction and increased Type I IFN signaling. These findings may have important implications for these high-risk patients. However, the findings are somewhat limited in

scope given the small sample size and inability to look at the relationship between immune dysfunction and clinical features of T2D.

Major comments:

- Introduction and discussion are somewhat limited in scope. They do not acknowledge the emerging literature that describes the effects of SARS-CoV-2 on metabolism and the relationship between glucose levels on regulating viral replication and cytokine production in monocytes (Campso Codo et al. Cell Metabolism 2020). In the discussion, the authors should also mention the potential relationship between glucose levels and modulation of IRF5 through O-GlcNAcylation.
- Based on methods, it looks like more information is available about the status of T2D in these patients. Do they have controlled or uncontrolled disease? This information should be added to Table 1. While the sample size may be too small to perform comparisons, this information may be helpful in understanding this information in this population. Have you examined associations between immune profiles and HbA1c?
- How does the age of individual confound these results, particularly for those with T2D? While the cohorts were matched for age, was there any relationship between age and severity of disease?
- Figure 1C includes patients with T2D without COVID-19. Who are these patients? Where do they come from? Are they matched to COVID-19 cohorts?
- Were monocyte and monocyte subset counts also evaluated using flow cytometry or was the data limited to % and MFI? While monocytopenia was observed in full blood counts, you cannot really determine whether or not a specific subset is depleted at the level of overall numbers. You can only discuss changes in relative frequencies across subsets.
- Correlation does not necessarily mean causation. For example, in discussion of IRF5 findings on page 7, the authors wrote: "Correlative analyses to monocyte phenotypic and functional markers revealed a positive correlation between IRF5 and HLA-DR, with no correlation to CD14, CD16 nor FSC (Fig. 3f and Fig. S3e). These data indicate that IRF5 does not directly regulate monocyte class switch nor morphological changes, however a dependent relationship exists between IRF5 and HLA-DR. IRF5 may therefore impact antigen presentation capacity or other functions associated to HLA-DR." However, you cannot specifically prove this. This statement should be moved from the results into the discussion and should be mentioned as a possible interpretation.

Minor comments:

- Define M and K the first time they are used.
- Data needs to be put in the context of what is "normal" for COVID-19 negative individuals with and without diabetes. What are the expected frequencies of monocyte subsets etc. Will be critical if these markers are to be have value as prognostic value.

Referee #3 (Remarks for Author):

Alzaid et al investigate the impact of T2DM on COVID-19 severity. Currently, little is known on underlying mechanism of how T2DM affects disease and clinical outcomes of COVID-19 infected pateints. This is a well performed and powered study that deserves rapid publication.

Following issues should be addressed.

1. Do uninfected T2DM pateints have monocytopenia specific to quiescent cells and a decreased frequency of cytotoxic lymphocytes?

2. There have been some indications that nicotine affects disease -- or may even protect. Is there an association of smoking with disease severity?

3. Given Nlrp3 inflammasome is involved in some aspects of T2DM mediated inflammation and potentially involved in COVID-19, what is the status of inflammasome dependent cytokines in T2DM COVID patients

Referee #1

Major comments

The data about the changes monocytes amounts and amounts of total PBMC per ml of blood in nondiabetic versus diabetic patients in response to COVID-19 have to be provided.

Thank you for this comment. The absolute counts per ml of PBMC and major leukocyte populations from clinical FBCs are provided in supplementary Table S1. PBMC, monocyte and monocyte subpopulation counts were also calculated based on flow cytometry data and are listed below and have been added in Supplementary Table S2. Importantly, PBMC and monocyte counts were coherent with the clinical FBC data and monocytopenia of CD14⁺ monocytes and specific to CD14^{Hi}CD16⁻ classical monocytes in patients with type-2 diabetes was confirmed.

Table S2. Flow cytometry-based counts of PBMCs, monocytes and monocyte subpopulations in non-diabetic (ND) and type-2 diabetic (T2D) patients with COVID-19

Leukocyte populations (10 ⁹ /L or 10 ⁶ /mL)	ND (n=15)	T2D (n=30)	p -value
CD45 ⁺ PBMCs	6.83 (3.71-13.44)	6.06 (2.35-15.49)	0,444
CD14 ⁺ Monocytes	0.75 (0.24-1.71)	0.49 (0.13-1.26)	*0,032
CD14 ^{Hi} CD16 ⁻ Classical	0.55 (0.33-1.01)	0.35 (0.08-1.11)	*0,005
CD14 ^{Hi} CD16 ⁺ Intermediate	0.08 (0.01-0.32)	0.07 (0.002-0.25)	0,660
CD14 ^{Lo} CD16 ⁻ Non-classical	0.02 (0.00-0.07)	0.01 (0.00-0.06)	0,054

**p*<0,05

IL8 concentrations have to be measured in the serum of patients. Do secreted levels of IL-8 directly correlate with gene expression in PBMC? Are circulating levels of IL8 biologically significant?

We thank the reviewer for raising this important point. We have quantified the levels of circulating IL-8 in plasma. We found that plasma IL8 was higher in T2D patients with COVID-19 than in non-T2D patients (Response Figure 1A.). However, we found no correlation between PBMC gene expression levels of IL-8 and circulating levels of IL-8 (Response Figure 1B.). This may indicate that over all a significant proportion of circulating IL-8 may originate from innate immune cells at the site of infection rather than from PBMC. Interestingly, when ND and T2D patients are separated, in ND patients IL8 plasma levels correlate positively to PBMC mRNA expression levels, whilst the correlation is negative in T2D, this further confirms an altered immune response to SARS-CoV-2 in the diabetic state. Circulating levels are biologically relevant with some patients above the physiological threshold (<5 pg/ml). Levels detected in our cohort are coherent with recent reports of plasma IL-8 in COVID-19 (Chi *et al* 2020 J Infect DOI: 10.1093/infdis/jiaa363; Gong *et al* 2020 medRxiv; DOI: 10.1101/2020.02.25.20025643). Plasma IL-8 concentrations between ND and T2D patients has been added to Supplementary Figure S2 to complement mRNA expression data in the main figures, this has also been added to the text of the results section.

Response Figure 1. Circulating levels of IL-8 in non-diabetic (ND) and type-2 diabetic (T2D) patients with COVID-19. A. IL-8 was measured in plasma from ND and T2D patients. B. Correlative analysis between IL8 mRNA expression in PBMCs from ND and T2D patients with COVID-19 and plasma IL-8 concentrations from the same patients. C. Correlative analyses separating ND and T2D patients with COVID-19. Data are presented as mean +/- SEM. Differences between groups were evaluated with unpaired t-test. For correlative analysis Spearman's test was carried out. * $p < 0.05$.

Which medications have been used for the diabetic patients? Which subgroups can be defined based on the treatment? How anti-diabetic medications affect monocytes subsets and morphology?

And

Which vascular complications have been found in diabetic patients? Do monocytes changes correlate with vascular complications depending on COVID-19 infection?

We thank the reviewer for these comments, although we wished to address these issues in our initial submission, several factors related to the hospital context of the COVID-19 pandemic limited the integrality of clinical data. Taking insulin treatment into consideration, we compared monocyte subsets according to the pre-hospitalisation use of insulin (which was systematically indicated) and observe no difference (Response Table 1).

Response Table 1. Insulin treatment and monocyte subset frequency in COVID-19 patients with T2D

Monocyte subset	Insulin (-)	Insulin (+)	p-value
Classical monocytes	4.63 +/- 3.09	4.87 +/- 2.50	0.527
Intermediate monocytes	1.17 +/- 1.13	1.09 +/- 1.24	0.306
Non-classical monocytes	0.15 +/- 0.25	0.20 +/- 0.25	0.114

On the other hand, vascular complications were rarely indicated in medical reports and were not screened for without indication, we were not able to carry out an analysis concerning this criterion. We did have prevalence of hypertension, BMI and age for all patients included, and we adjusted for these vascular risk factors with respect to immunophenotypic data. Adjusting is detailed in the methods section and in the MANOVA/ANOVA Supplementary Table S4. Sections relevant to these adjustments have been highlighted in the revised manuscript.

Minor comments

It is recommended to make more straight forward formulations in the abstract listing what are the major differences in monocytes phenotype identified in non-diabetic versus diabetic patients in response to COVID-19 infection

We thank the reviewer for this comment. We've rewritten the abstract section addressing results to make more direct statements of our major findings. The rewritten section is highlighted in the abstract of the revised manuscript as:

“Lymphocytopenia and specific loss of cytotoxic CD8⁺ lymphocytes was associated with severe COVID-19 and requirement for intensive care in both non-diabetic and T2D patients. A morphological anomaly of increased monocyte size and monocytopenia restricted to classical CD14^{Hi} CD16⁺ monocytes were specifically associated with severe COVID-19 in patients with T2D requiring intensive care. Increased expression of inflammatory markers reminiscent of the type-1 interferon pathway (INFB1, IL6, CCL2) underlaid the immunophenotype associated with T2D. These immunophenotypic and hyperinflammatory changes may contribute to increased voracity of COVID-19 in T2D.”

Fold change for all measured parameters has to be indicated in the text of Results. The biological significance of the fold change has to be commented.

Fold changes were calculated and have been added in the text of the results section.

Referee #2 (Remarks for Author):

Major comments:

Introduction and discussion are somewhat limited in scope. They do not acknowledge the emerging literature that describes the effects of SARS-CoV-2 on metabolism and the relationship between glucose levels on regulating viral replication and cytokine production in monocytes (Campso Codo et al. Cell Metabolism 2020).

We thank the reviewer for these suggestions, we have expanded the introduction to detail the following important studies relevant to T2D and COVID-19:

1. Zhu, L., She, Z. G., Cheng, X... Guo, J., Zhang, B. H., and Li, H. (2020) Association of Blood Glucose Control and Outcomes in Patients with COVID-19 and Pre-existing Type 2 Diabetes. *Cell Metab* **31**, 1068-1077 e1063
2. Wang, Z., Du, Z., and Zhu, F. (2020) Glycosylated hemoglobin is associated with systemic inflammation, hypercoagulability, and prognosis of COVID-19 patients. *Diabetes Res Clin Pract* **164**, 108214
3. Codo, A. C., Davanzo, G. G., Monteiro, L. B... Nakaya, H. I., Farias, A. S., and Moraes-Vieira, P. M. (2020) Elevated Glucose Levels Favor SARS-CoV-2 Infection and Monocyte Response through a HIF-1alpha/Glycolysis-Dependent Axis. *Cell Metab*
4. Wang, Q., Fang, P., He, R...Peng, G., Rao, L., Liu, S. (2020) O-GlcNAc transferase promotes influenza A virus-induced cytokine storm by targeting interferon regulatory factor-5. *Sci Adv*

Accordingly, we have updated parts of the discussion as highlighted in the text and added this section to the introduction:

“Key reports have found important links between systemic metabolism, glucose homeostasis and responses to COVID-19. Notably, studies have shown that glycaemic variability strongly influences outcome in COVID-19, where poorly-controlled blood glucose was associated with markedly higher mortality compared to patients with well-controlled blood glucose (1). Similarly, high glycosylated haemoglobin (HbA1c), a proxy of glycaemic instability, has been associated with low oxygen saturation, inflammation and hypercoagulability in patients with COVID-19 (2). More recently, mechanistic studies have shown that the elevated glucose that sustains inflammatory metabolism in immune cells directly promotes viral replication and cytokine production in SARS-CoV-2 infection (3).”

In the discussion, the authors should also mention the potential relationship between glucose levels and modulation of IRF5 through O-GlcNAcylation.

Thank you for this very interesting comment, indeed two very relevant studies by Wang *et al* and Kim *et al* make important mechanistic advances in linking hyperglycaemia to cellular metabolism and the consequent increased activation and dysregulated expression of IRF5 in viral infection and in T2D (Wang *et al* 2020 Sci Adv DOI: 10.1126/sciadv.aaz7086; Kim *et al* 2017 Cell Rep DOI: 10.1016/j.celrep.2017.06.088). These articles' relevance to SARS-CoV-2 infection and severity in T2D is confirmed by their further citation in a recently published opinion article in Front Endocrinol (Laviada-Molina *et al* 2020 doi: 10.3389/fendo.2020.00514). Accordingly, we have commented on findings from these reports in the discussion:

“The molecular mechanisms linking hyperglycaemia and cellular glucose metabolism directly to an IRF5-dependent cytokine storm have recently been described in the case of influenza A virus (IAV) infection, of which some mechanisms may be shared with COVID-19 (4). Indeed, increased glucose consumption is characteristic of inflammatory effector function of macrophages, where glucose shuttling to the hexosamine biosynthesis pathway provides a substrate for O-GlcNAcylation of IRF5 on serine-430 and its subsequent K63-linked ubiquitination. These posttranslational modifications allow the downstream processing of IRF5 and the engagement of its pro-inflammatory transcriptional activities. IRF5 O-GlcNAcylation in human PBMC and subsequently increased IL8 and IL6 levels in circulation were associated with increased blood glucose in IAV infected patients (4). The IRF5 expression and cytokine profile reported in IAV infection are similar to what we observe in PBMC from SARS-CoV-2 infected patients, comforting our hypothesis that similar mechanisms are at play. Importantly, the K63-linked ubiquitination of IRF5, required for its nuclear translocation, is an indispensable mechanism in macrophages that mediate metabolic inflammation and loss of glycaemic homeostasis in T2D (independently of viral infection) (5). A recent report has also dissociated IRF5-mediated cytokine production from an inhibitory effect on viral replication in IAV. These studies, and a recent pre-print implicating impaired type-1 interferon signalling in severity of COVID-19 cases, strongly support a key role for IRF5 and the type-1 interferon response in increased severity in T2D (6,7).

Taken together, our data and previous reports indicate that basal levels of IRF5, preceding SARS-CoV-2 infection, are dysregulated in T2D patients, our supposition is comforted by a recent opinion article describing the hypothetical molecular mechanisms (8). Monocytopenia and rapid class switch of monocytes in T2D with COVID-19, may be the result of an exuberant viral response from an immune system primed on an inflammatory background. IRF5-linked hyperinflammation will induce eager damage-seeking behaviour, antigen presentation and cytokine release, without affecting viral replication. Thus, contributing to the cytokine storm syndrome that characterises severe COVID-19 (9).”

Based on methods, it looks like more information is available about the status of T2D in these patients. Do they have controlled or uncontrolled disease? This information should be added to Table 1. While the sample size may be too small to perform comparisons, this information may be helpful in understanding this information in this population. Have you examined associations between immune profiles and HbA1c?

and

How does the age of individual confound these results, particularly for those with T2D? While the cohorts were matched for age, was there any relationship between age and severity of disease?

HbA1c is indeed a good indicator of controlled versus uncontrolled disease. The median (IQR) for HbA1c in this T2D cohort was 7.8 % (7.3-10.3). In more detail, 41 % of patients had uncontrolled diabetes with HbA1c greater than 8 %, and 28 % had HbA1c greater than 10 %, almost all of these corresponding to unknown diabetes that was fortuitously discovered upon their admission for COVID-19.

Taking age into account and anticipating its possible effect as well as that of HbA1c, we had matched uninfected T2D patients to patients with T2D and COVID-19; we had also age-matched ND patients with COVID-19. Age, amongst other criteria, was adjusted for when analysing patient immunophenotype (MANOVA/ANOVA in Supplementary Table S4). The relevant sections have been highlighted in the revised manuscript. In order to respond as fully as possible to this comment we have produced a correlation matrix of all the immunophenotypic and clinical data we had procured, and we confirm no major correlations of age nor HbA1c to clinical criteria nor to immunophenotypic results in this cohort (Response Figure 2).

Response Figure 2. Correlation matrices of all clinical and immunophenotypic data in COVID-19 patients with or without T2D. R- and p-values represented by grayscale bar.

Figure 1C includes patients with T2D without COVID-19. Who are these patients? Where do they come from? Are they matched to COVID-19 cohorts?

Non-COVID-19 patients with T2D in this study were recruited as part of an observational study (NCT02671864) that was on-going in the same clinical service prior to the COVID-19 pandemic. Data included in this study was used from T2D patients without COVID-19 matched to those recruited as part of the COVID-19 cohort. Patients were matched for age, gender, BMI and for hypertension and HbA1c to T2D patients with COVID-19. A statement accounting for non-COVID-19 patients has been added to the *Material and Methods* section under the *Human Populations* subsection:

“Non-COVID-19 T2D patients were included from the same clinical service prior to the COVID-19 pandemic as part of an on-going observational study (NCT02671864), patients were matched to the T2D COVID-19 cohort in terms of age, gender, BMI, hypertension and HbA1c.”

Were monocyte and monocyte subset counts also evaluated using flow cytometry or was the data limited to % and MFI? While monocytopenia was observed in full blood counts, you cannot really determine whether or not a specific subset is depleted at the level of overall numbers. You can only discuss changes in relative frequencies across subsets.

We thank the reviewer for bringing up this important point. In our study the main method of immunophenotypic evaluation was by flow cytometry with CD45 as a haematopoietic lineage marker; CD14, CD3, CD20, CD56 as monocyte, lymphocyte, B-cell and NK-cell lineage markers, respectively. Further activation and phenotypic markers included CD16, HLA-DR, CD123, CD11c, CD4 and CD8. These markers were chosen based on the Immunological Genome (ImmGen) classification of immune cell phenotypes (Heng et al 2008 Nat Immunol; DOI: 10.1038/ni1008-1091). The gating strategy applied (Figure S1a in manuscript and Response Figure .3) with the use of such markers has been adapted from previously published analyses of high-dimensional flow cytometry data (Melzer et al 2015 Cytometry B Clin Cytom DOI: 10.1002/cyto.b.21234; Autissier et al 2010 J Immunol Methods doi:10.1016/j.jim.2010.06.017). The corresponding sections have been highlighted in the manuscript main text and methods section.

Response Figure 3. Markers and gating strategy applied for in flow cytometric phenotyping of immune populations in peripheral circulation. Flow cytometry gating to quantify immune cell populations and subpopulations in human PBMCs

We have quoted frequencies in figures and in the text of the results section in line with reporting conventions for immunophenotypic analyses. This form of reporting allows correcting for cytopenia in parent populations that may mask changes in subpopulations. As a further confirmation, we applied an unsupervised analysis to marker expression data from all events analysed by flow cytometry, the tSNE algorithm resulted in population groupings that confirmed appropriate gating and interpretation of cell frequency expression data (Figure 4d). Comforting our findings and adding clinical relevance, clinical laboratory FBCs corroborated lymphocytopenia in ND and T2D patients with COVID-19 and monocytopenia in T2D patients with COVID-19 (Table S1). To address whether immunophenotyping data could determine a specific subset depletion in overall numbers, we have added absolute counts calculated from flow cytometric immunophenotyping of PBMCs, monocytes and monocyte subsets in Table S2 in the revised manuscript.

Correlation does not necessarily mean causation. For example, in discussion of IRF5 findings on page 7, the authors wrote: "Correlative analyses to monocyte phenotypic and functional markers revealed a positive correlation between IRF5 and HLA-DR, with no correlation to CD14, CD16 nor FSC (Fig. 3f and Fig. S3e). These data indicate that IRF5 does not directly regulate monocyte class switch nor morphological changes, however a dependent relationship exists between IRF5 and HLA-DR. IRF5 may therefore impact antigen presentation capacity or other functions associated to HLA-DR." However, you cannot specifically prove this. This statement should be moved from the results into the discussion and should be mentioned as a possible interpretation.

We thank the reviewer for this comment, the phrasing and interpretation of our results is better suited to the discussion section than the results section. Modified sentence has been highlighted in the results section and the following phrase has been added to the discussion:

"Whilst IRF5 expression did not correlate to monocyte activation markers, we did observe a positive correlation between IRF5 and HLA-DR. A positive correlation indicates a possible dependent relationship where HLA-DR may form part of a mechanisms feeding back to increase IRF5 expression, alternatively IRF5 may impact monocyte antigen presentation capacity or other HLA-DR associated functions."

Minor comments:

Define M and K the first time they are used.

We have replaced "M" for "million" and "K" with "thousand" in the main text.

Data needs to be put in the context of what is "normal" for COVID-19 negative individuals with and without diabetes. What are the expected frequencies of monocyte subsets etc. Will be critical if these markers are to be have value as prognostic value.

We thank the reviewer for this observation. We have updated Table S1 with reference ranges for leukocytes, neutrophils, lymphocytes and monocytes in clinical full blood counts to put the effects of COVID-19 in the context of healthy individuals.

Table S1. Full blood count of COVID-19 patients in the non-diabetic (ND) and type-2 diabetic (T2D) groups at admission to hospital

Cell counted (10 ⁹ /L or 10 ⁶ /mL)	ND (n=15)	T2D (n=30)	p-value	Reference ranges (10 ⁹ /L or 10 ⁶ /mL)
Leukocytes	6.3 (5.5-7.8)	6 (4.8-7.45)	0.255	4.0 – 11.0
Neutrophils	3.93 (3.23-5.27)	3.81 (2.92-5.08)	0.497	2.0 - 7.5
Lymphocytes	1.18 (1.07-1.75)	1.3 (0.91-1.65)	0.860	1.5 - 4.5
Monocytes	0.63 (0.48-0.76)	0.44 (0.33-0.62)	0.028	0.2 - 0.8

With regards to flow cytometric immunophenotyping we added Table S3 with mean frequency and standard deviation of each population and subpopulation quantified, per category of patient including values from healthy donors. Healthy donors were recruited from the national blood bank. We could not include healthy donor data as part of the main manuscript, main results or analyses as we do not have access to demographic, physical or clinical data and thus cannot match them to the COVID-19 cohort in terms of age, gender nor other criteria. We also cannot

confirm that the time between blood drawing and analysis is the same for healthy donors as for COVID-19 and uninfected T2D patients having frequented the hospital. Nonetheless, trends can be inferred from healthy donor data where we confirm that lymphopenia is characteristic of COVID-19 infection, including decreased mean frequency of CD8+ lymphocytes (6.4% and 8.8% in ND and T2D uninfected patients versus 3.7% and 5.0% in ND and T2D COVID-19 patients). We confirm relative monocytopenia is characteristic of COVID-19 in patients with T2D (9.0% and 9.9% in ND and T2D uninfected patients versus 9.9% and 7.6% in ND and T2D COVID-19 patients); similarly this is specific to CD14-Hi CD16- classical monocytes (7.4% and 8.3% in ND and T2D uninfected patients versus 7.9% and 5.7% in ND and T2D COVID-19 patients). Values in healthy donors are coherent with reports having applied similar immunophenotyping approaches (Melzer *et al* 2015 Cytometry B doi: 10.1002 /cyto.b.21234; Autissier *et al* 2010 J Immunol Methods doi: 10.1016/j.jim.2010.06.017).

Table S3. Flow cytometry-based frequency of immune populations and subpopulations in non-diabetic (ND) and type-2 diabetic (T2D) patients with and without COVID-19

Immune population (mean freq +/- SD)	COVID-19		Uninfected	
	ND (n=15)	T2D (n=30)	ND (n=36)	T2D (n=22)
Lymphocytes	10.8 +/- 6.1	12.2 +/- 10.1	22.2 +/- 9.5	31.1 +/- 18.4
CD8+	3.7 +/- 2.4	5.0 +/- 5.0	6.4 +/- 4.5	8.8 +/- 7.3
CD4+	6.8 +/- 5.1	6.1 +/- 4.4	10.4 +/- 5.0	13.5 +/- 15.0
DN	0.9 +/- 0.7	0.8 +/- 0.7	2.3 +/- 2.3	1.5 +/- 2.1
DP	0.1 +/- 0.1	0.07 +/- 0.07	0.2 +/- 0.1	7.3 +/- 10.1
Monocytes	9.9 +/- 2.0	7.6 +/- 3.6	9.0 +/- 2.2	9.9 +/- 4.9
Classical	7.9 +/- 3.6	5.7 +/- 2.5	7.4 +/- 2.7	8.3 +/- 6.7
intermediate	1.0 +/- 1.3	1.2 +/- 1.5	1.5 +/- 0.9	1.9 +/- 1.7
non-classical	0.3 +/- 0.3	0.2 +/- 0.2	0.4 +/- 0.3	0.2 +/- 0.4
DCs	1.5 +/- 0.9	1.7 +/- 0.9	0.9 +/- 0.4	1.9 +/- 2.1
mDCs	0.2 +/- 0.2	0.4 +/- 0.3	0.3 +/- 0.3	0.2 +/- 0.3
pDCs	0.3 +/- 0.5	0.2 +/- 0.3	0.3 +/- 0.2	0.3 +/- 0.5
B-cells	2.5 +/- 2.8	2.4 +/- 2.0	5.2 +/- 9.1	5.8 +/- 5.1
NK-cells	4.9 +/- 2.5	2.4 +/- 2.0	5.2 +/- 9.1	5.8 +/- 5.1
iNK cells	3.7 +/- 2.3	4.4 +/- 2.7	3.8 +/- 5.3	2.8 +/- 3.0
NKT- cells	1.1 +/- 1.3	1.7 +/- 1.7	0.9 +/- 0.9	1.4 +/- 1.5
Granulocytes	62 +/- 12.7	59.2 +/- 14.8	37.9 +/- 16.4	43.9 +/- 17.7

Referee #3 (Remarks for Author):

Do uninfected T2DM patients have monocytopenia specific to quiescent cells and a decreased frequency of cytotoxic lymphocytes?

Thank you for this interesting question, uninfected T2D patients did not have lymphopenia nor monocytopenia in the parent immune populations nor in CD8+ cytotoxic lymphocytes and quiescent classical monocytes (Figure S1b in the revised manuscript and excerpt in Response Figure 4). We confirm that lymphocyte and monocyte loss in patients with T2D and COVID-19 is due to comorbidity of both conditions and not due to diabetes *per se*. To place these results in a wider context in the revised manuscript we have added reference ranges to clinical FBCs (Table S1) and included immunophenotyping data from uninfected ND healthy donors (Table S3). These tables are also included above in this response to reviewers' comments.

Response Figure 4. Lymphocyte, CD8+ cytotoxic lymphocyte, monocyte and quiescent classical monocytes frequencies in patients with T2D and in COVID-19 patients with and without T2D.

There have been some indications that nicotine affects disease -- or may even protect. Is there an association of smoking with disease severity?

Unfortunately, smoking status was known for only 31 patients, of which only 3 were smokers, making it impossible to carry out more precise analyses.

Given Nlrp3 inflammasome is involved in some aspects of T2DM mediated inflammation and potentially involved in COVID-19, what is the status of inflammasome dependent cytokines in T2DM COVID patients

We thank the reviewer for this very insightful comment. We initially hypothesised that the NLRP3 inflammasome and/or Interferon signalling played important roles in the hyperinflammation and severity of COVID-19 in patients with T2D. PBMC gene expression of IL1B, the major NLRP3 inflammasome end-product, was increased with COVID-19 infection, but invariant between ND COVID-19 patients and COVID-19 patients with T2D (Response Figure 5A). Whilst IRF5 and IFNB1 that form the type-1 interferon response were increased in COVID-19 and further increased in COVID-19 with pre-existing T2D (Response Figure 5B). The important role of interferon signalling in T2D and COVID-19 comorbidity is strongly supported by these data as well as the rapid accumulation of research reports demonstrating that dysregulated interferon signalling forms a major part of COVID-19 pathology (Hadjadj *et al* 2020 Science DOI: 10.1126/science.abc6027; Sawalha *et al* 2020 Clin Immunol; Huang *et al* 2020 MedRxiv DOI: 10.1101/2020.03.15.20033472; Zhou *et al* 2020 Front Immunol DOI: 10.3389/fimmu.2020.01061; Acharya *et al* 2020 Nat Rev Immunol DOI: 10.1038/s41577-020-0346-x). Accordingly, we have included our data on IL1B expression in Supplementary Figure S2f and added sections in the results and discussion to address this point.

Response Figure 5. IL1B, IRF5 and IFNB1 gene expression in PBMC from uninfected patients with T2D and from COVID-19 patients with T2D and without (ND).

References

1. Zhu, L., She, Z. G., Cheng, X., Qin, J. J., Zhang, X. J., Cai, J., Lei, F., Wang, H., Xie, J., Wang, W., Li, H., Zhang, P., Song, X., Chen, X., Xiang, M., Zhang, C., Bai, L., Xiang, D., Chen, M. M., Liu, Y., Yan, Y., Liu, M., Mao, W., Zou, J., Liu, L., Chen, G., Luo, P., Xiao, B., Zhang, C., Zhang, Z., Lu, Z., Wang, J., Lu, H., Xia, X., Wang, D., Liao, X., Peng, G., Ye, P., Yang, J., Yuan, Y., Huang, X., Guo, J., Zhang, B. H., and Li, H. (2020) Association of Blood Glucose Control and Outcomes in Patients with COVID-19 and Pre-existing Type 2 Diabetes. *Cell Metab* **31**, 1068-1077 e1063
2. Wang, Z., Du, Z., and Zhu, F. (2020) Glycosylated hemoglobin is associated with systemic inflammation, hypercoagulability, and prognosis of COVID-19 patients. *Diabetes Res Clin Pract* **164**, 108214
3. Codo, A. C., Davanzo, G. G., Monteiro, L. B., de Souza, G. F., Muraro, S. P., Virgilio-da-Silva, J. V., Prodonoff, J. S., Carregari, V. C., de Biagi Junior, C. A. O., Crunfli, F., Jimenez Restrepo, J. L., Vendramini, P. H., Reis-de-Oliveira, G., Bispo Dos Santos, K., Toledo-Teixeira, D. A., Parise, P. L., Martini, M. C., Marques, R. E., Carmo, H. R., Borin, A., Coimbra, L. D., Boldrini, V. O., Brunetti, N. S., Vieira, A. S., Mansour, E., Ulaf, R. G., Bernardes, A. F., Nunes, T. A., Ribeiro, L. C., Palma, A. C., Agrela, M. V., Moretti, M. L., Sposito, A. C., Pereira, F. B., Velloso, L. A., Vinolo, M. A. R., Damasio, A., Proenca-Modena, J. L., Carvalho, R. F., Mori, M. A., Martins-de-Souza, D., Nakaya, H. I., Farias, A. S., and Moraes-Vieira, P. M. (2020) Elevated Glucose Levels Favor SARS-CoV-2 Infection and Monocyte Response through a HIF-1alpha/Glycolysis-Dependent Axis. *Cell Metab*
4. Wang, Q., Fang, P., He, R., Li, M., Yu, H., Zhou, L., Yi, Y., Wang, F., Rong, Y., Zhang, Y., Chen, A., Peng, N., Lin, Y., Lu, M., Zhu, Y., Peng, G., Rao, L., and Liu, S. (2020) O-GlcNAc transferase promotes influenza A virus-induced cytokine storm by targeting interferon regulatory factor-5. *Sci Adv* **6**, eaaz7086
5. Kim, D., Lee, H., Koh, J., Ko, J. S., Yoon, B. R., Jeon, Y. K., Cho, Y. M., Kim, T. H., Suh, Y. S., Lee, H. J., Yang, H. K., Park, K. S., Kim, H. Y., Lee, C. W., Lee, W. W., and Chung, D. H. (2017) Cytosolic Pellino-1-Mediated K63-Linked Ubiquitination of IRF5 in M1 Macrophages Regulates Glucose Intolerance in Obesity. *Cell Rep* **20**, 832-845
6. Forbester, J. L., Clement, M., Wellington, D., Yeung, A., Dimonte, S., Marsden, M., Chapman, L., Coomber, E. L., Tolley, C., Lees, E., Hale, C., Clare, S., Udalova, I., Dong, T., Dougan, G., and Humphreys, I. R. (2020) IRF5 Promotes Influenza Virus-Induced Inflammatory Responses in Human Induced Pluripotent Stem Cell-Derived Myeloid Cells and Murine Models. *J Virol* **94**
7. Hadjadj, J., Yatim, N., Barnabei, L., Corneau, A., Boussier, J., Pere, H., Charbit, B., Bondet, V., Chenevier-Gobeaux, C., Breillat, P., Carlier, N., Gauzit, R., Morbieu, C., Pene, F., Marin, N., Roche, N., Szwebel, T. A., Smith, N., Merklings, S., Treluyer, J. M., Veyer, D., Mouthon, L., Blanc, C., Tharaux, P. L., Rozenberg, F., Fischer, A., Duffy, D., Rieux-Laucat, F., Kerneis, S., and B., T. (2020) Impaired type I interferon activity and exacerbated inflammatory responses in severe Covid-19 patients. *medRxiv* doi: <https://doi.org/10.1101/2020.04.19.20068015>
8. Laviada-Molina, H. A., Leal-Berumen, I., Rodriguez-Ayala, E., and Bastarrachea, R. A. (2020) Working Hypothesis for Glucose Metabolism and SARS-CoV-2 Replication: Interplay Between the Hexosamine Pathway and Interferon RF5 Triggering Hyperinflammation. Role of BCG Vaccine? *Frontiers in Endocrinology* **11**
9. Gruber, C. (2020) Impaired interferon signature in severe COVID-19. *Nat Rev Immunol*

14th Aug 2020

Dear Dr. Alzaid,

Thank you for the submission of your revised manuscript to EMBO Molecular Medicine. We have now received the enclosed report from the referee who was asked to re-assess it. As you will see the reviewer is now supportive and I am pleased to inform you that we will be able to accept your manuscript pending final editorial amendments.

Please submit your revised manuscript within two weeks. I look forward to seeing a revised form of your manuscript as soon as possible.

Yours sincerely,

Celine Carret

Celine Carret, PhD
Senior Editor
EMBO Molecular Medicine

*** Instructions to submit your revised manuscript ***

***** Reviewer's comments *****

Referee #2 (Remarks for Author):

The authors carefully considered the comments from the reviewers and made the suggested changes. The manuscript is now suitable for publication.

The authors performed the requested editorial changes.

19th Aug 2020

Dear Dr. Alzaid,

We are pleased to inform you that your manuscript is accepted for publication and is now being sent to our publisher to be included in the next available issue of EMBO Molecular Medicine.

Please make sure that Dr. Gautier update his profile in our system to link his ORCID iD as we cannot do that for him (data protection law). We won't be able to send the article to our publisher without it!

Please read below for additional IMPORTANT information regarding your article, its publication and the production process.

Congratulations on your interesting work,

Celine Carret

Celine Carret, PhD
Senior Editor
EMBO Molecular Medicine

Follow us on Twitter @EmboMolMed
Sign up for eTOCs at embopress.org/alertsfeeds

Corresponding Author Name: Fawaz Alzaid

Journal Submitted to: EMBO Mol Med

Manuscript Number: EMM-2020-13038